# Multivariate information theory uncovers synergistic subsystems of the human cerebral cortex

Thomas F. Varley [1,2,4✉], Maria Pope[1,3,4], Joshua Faskowitz[2,3] & Olaf Sporns[1,2,3]

One of the most well-established tools for modeling the brain is the functional connectivity network, which is constructed from pairs of interacting brain regions. While powerful, the network model is limited by the restriction that only pairwise dependencies are considered and potentially higher-order structures are missed. Here, we explore how multivariate information theory reveals higher-order dependencies in the human brain. We begin with a mathematical analysis of the O-information, showing analytically and numerically how it is related to previously established information theoretic measures of complexity. We then apply the O-information to brain data, showing that synergistic subsystems are widespread in the human brain. Highly synergistic subsystems typically sit between canonical functional networks, and may serve an integrative role. We then use simulated annealing to find maximally synergistic subsystems, finding that such systems typically comprise ≈10 brain regions, recruited from multiple canonical brain systems. Though ubiquitous, highly synergistic subsystems are invisible when considering pairwise functional connectivity, suggesting that higher-order dependencies form a kind of shadow structure that has been unrecognized by established network-based analyses. We assert that higher-order interactions in the brain represent an under-explored space that, accessible with tools of multivariate information theory, may offer novel scientific insights.

[1] School of Informatics, Computing & Engineering, Indiana University, Bloomington, IN 47405, USA. [2] Department of Psychological & Brain Sciences, Indiana University, Bloomington, IN 47405, USA. [3] Program in Neuroscience, Indiana University, Bloomington, IN 47405, USA. [4] These authors contributed equally: Thomas F. Varley, Maria Pope. ✉email: tvarley@iu.edu

Perhaps the most ubiquitous model used in complex systems is the network, which represents pairwise interactions between different elements of a system as directed or undirected graphs[1,2]. While network models can be extremely powerful, they are also fundamentally limited by the rule that every interaction between elements is strictly bivariate. Hence, interactions between three or more nodes must be indirectly inferred, using methods such as motifs[3], transitivity or clustering coefficients[4], and mapping cores or mesoscale communities[5,6]. Increasingly, statistical interactions involving more than two elements (termed higher-order interactions) are recognized as being a key feature of complex systems[7,8]. This makes the task of recognizing and modeling higher-order structures an important, developing field. However, a lack of well-developed, formal tools, as well as the inherent computational and combinatorial difficulties associated with higher-order interactions, have limited their application. In neuroscience, higher-order interactions have been theoretically implicated as building blocks of complexity[9,10] and functional integration[11]. Empirically, they have been found at multiple scales, including in neuronal networks[12–17], electrophysiological signals[18,19], and fMRI BOLD data[20–22], where higher-order interactions have been proposed to relate to emergent mental phenomena and consciousness[23].

Recently, Rosas and Mediano[24] proposed that information theory could be used to identify higher-order interactions in multivariate systems, and furthermore, that it is possible to disentangle qualitatively different kinds of interactions, characterized by pairwise redundant and synergistic modes of information sharing. Intuitively, redundant information corresponds to information that is copied over many different elements such that the observation of a single element resolves the corresponding uncertainty in all of the other elements. In contrast, synergistic information sharing occurs when uncertainty can only be resolved by considering the joint state of two or more variables. This space of redundant and synergistic interactions in the brain remains largely unexplored, as it comprises interactions that are typically inaccessible to a bivariate, functional connectivity network analysis. Synergy is of potential interest because it tracks the ability of the brain to generate novel information through the interactions of multiple brain regions (sometimes called information modification)[25]. In studies of cortical neural networks, synergy has been associated with neural computation (the genesis of new information through a non-trivial interaction of multiple inputs)[12–15,17,26].

Much of the previous work on higher-order information in neuroscience has used the partial information decomposition (PID) framework[27,28], which provides a complete decomposition of the joint mutual information into atomic information components. While powerful, the PID framework has some fairly strict limitations that have hindered its adoption by the wider complex systems community. The first is that it requires partitioning a system into sources and targets, and does not allow analysis of the whole system qua itself. The second is that, due to the combinatorial explosion of information atoms, analysis of more than five or six elements is impossible. Given that even small systems can have hundreds, or even thousands of elements, this is a severe limitation. Finally, the PID is unusual in that, while it reveals the structure of multivariate information, actually calculating values from data requires an additional step: the selection of a redundancy function that quantifies some notion of redundant information. This is a surprisingly difficult task, as many redundancy functions have been proposed, and different choices can lead to radically different descriptions of the same system[29,30].

Rosas et al. introduced the O-information[24] as an alternative measure, which gives an overall estimate of the extent to which a system is redundancy dominated or synergy dominated, without requiring the incredible computational cost or ad hoc choices required by the PID. The O-information reveals the global structure of the information-sharing dependencies in a system. Negative O-information indicates the presence of predominantly synergistic interactions, while positive O-information indicates predominately redundant interactions. Despite its strong appeal as a quantitative metric, the origins and neural manifestations of O-information have remained elusive, if not enigmatic[31].

In this work, we apply a range of information-theoretic measures to resting state fMRI data acquired from human cerebral cortex with the aim of identifying ensembles of regions (subsystems) that express specific modes of higher-order statistical dependencies. First, we introduce the mathematical machinery required to derive the O-information, and its interpretation in the context of multivariate information sharing processes. We derive an analytic relationship betwee the O-information and other, more well-known, multivariate metrics such as the Tononi-Sporns-Edelman complexity[10]. Then we apply multivariate information metrics to brain data and uncover the presence of abundant and widely distributed subsystems expressing synergy (negative O-information) across the entire cerebral cortex. Finally, we discuss what our insights reveal about the structure and functional roles of higher-order relations in brain activity.

## Results
### Theory
*Integration, segregation, redundancy, synergy.* A fundamental idea in modern theoretical neuroscience states that the nervous system maintains a balance between integration and segregation[9]. The integration-segregation balance principle is based on the insight that the nervous system combines regional elements of functional specialization, with system-wide functional integration. Considerable empirical work has gone into the neural integration-segregation hypothesis, and the on-going balance of integrated and segregated dynamics has been found to be regulated by distinct neuromodulatory systems[32,33], and correlates with conscious awareness[34–38].

The segregation-integration spectrum is typically visualized as a one-dimensional space: on one extreme the system is totally disintegrated and every element is behaving entirely independently of all the others. On the other extreme is the case of total integration: every element synchronizes with every other element so that the whole system is densely connected. In the middle there is a complex regime where the system combines elements of independence and integration. As it was originally formulated, integration and segregation were discussed in the contexts of networks, and higher-order interactions were inferred via partitioning the system into subsets of varying numbers of nodes[9]. These arguments pre-dated the rigorous, mathematical distinction between redundancy and synergy, introduced in the work of Williams and Beer almost two decades later[27]. Building on these foundations, as well as the definition of O-information from Rosas et al.[24], we argue that the notion of integration can be expanded to include redundant integration and synergistic integration. The result is a rich space described by distinct dimensions of integration, segregation, redundancy, and synergy (although these do not form an orthogonal basis). This high-dimensional, qualitative configuration space may be viewed as an informational morphospace[39–41] and provides a framework for the detailed comparison of different systems.

*Information theory and higher-order information-sharing.* In this section, we introduce the basics of information theory necessary to understand its application to higher-order relationships. For a

more thorough introduction, readers may be interested in Cover & Thomas[42]. The basic object of study in information theory is the entropy[43], which quantifies the uncertainty that we, as observers, have about the state of a variable $X$. If the states of $X$ are drawn according to the probability distribution $P(X = x)$ with Support Set $\mathcal{X}$, then the entropy of $X$ is:

$$H(X) = -\sum_{x \in \mathcal{X}} P(x) \log_2 P(x) \qquad (1)$$

This classic formulation of entropy assumes that $X$ is a discrete random variable, although for continuous data, the generalization to differential entropy is reasonably straightforward (see Sec. Gaussian Information Theory).

Now consider two variables $X_1$ and $X_2$: how does knowing the state of $X_1$ reduce our uncertainty (the entropy) about the state of $X_2$? The answer is given by the mutual information[43], which can be written in two mathematically equivalent forms:

$$I(X_1; X_2) = H(X_1) + H(X_2) - H(X_1, X_2) \qquad (2)$$

$$= H(X_1, X_2) - [H(X_1 | X_2) + H(X_2 | X_1)] \qquad (3)$$

The bivariate mutual information is often applied in the study of complex systems for the inference of functional connectivity networks (e.g., refs. [44–48]), which can reveal the structure of dyadic interactions between different elements[49]. While functional connectivity networks are extremely powerful, they are fundamentally limited by their pairwise structure and are insensitive to higher-order interactions between three or more variables.

The natural place to begin an analysis of higher-order structures in neural data, then, is by attempting to generalize the mutual information to account for more than two variables. Unfortunately, there is no single unique generalization, and at least three are known to exist: the total correlation, the dual total correlation, and the interaction/co-information (which we will not explore in detail here)[42]. The total correlation (also referred to as the integration in ref. [9]), is formally a straightforward generalization of Eq. (2):

$$\text{TC}(\mathbf{X}) = \sum_{i=1}^{N} H(X_i) - H(\mathbf{X}) \qquad (4)$$

$$= D_{KL}(P(X_1, \dots, X_N)) || \prod_{i=1}^{N} P(X_i)) \qquad (5)$$

where $\mathbf{X}$ is a macro-variable comprised of an ensemble of multiple random variables: $\mathbf{X} = \{X_1, X_2, \dots, X_N\}$ and $D_{KL}()$ is the Kullback-Leibler divergence from prior distribution $Q(x)$ to posterior distribution $P(x)$:

$$D_{KL}(P || Q) = \sum_{x \in \mathcal{X}} P(x) \log \frac{P(x)}{Q(x)} \qquad (6)$$

The total correlation is low when every variable is independent, and high when every variable is individually highly entropic but the joint-state of the whole has low entropy. This occurs when the whole system is dominated by redundant interactions: the state of a single variable discloses a large amount of information about the state of every other variable.

The second generalization of mutual information is the dual total correlation, formally a generalization of Eq. (3):

$$\text{DTC}(\mathbf{X}) = H(\mathbf{X}) - \sum_{i=1}^{N} H(X_i | \mathbf{X}^{-i}) \qquad (7)$$

where $H(X_i | \mathbf{X}^i)$ refers to the residual entropy[50]: the uncertainty intrinsic to the the $i^{th}$ element of $\mathbf{X}$ that is not resolved by any other variable, or collection of variables, in $\mathbf{X}$. The difference between the joint entropy and the sum of the residual entropies is

all the entropy that is shared between at least two elements of $\mathbf{X}$ (i.e., is redundantly common to two or more elements). Curiously, while total correlation monotonically increases as $\mathbf{X}$ transitions from randomness to synchrony, the dual total correlation is low both for totally random, and totally synchronized systems, peaking when $\mathbf{X}$ is dominated by shared information.

Rosas et al.[24], propose that the difference between TC($\mathbf{X}$) and DTC($\mathbf{X}$) (first explored by James and Crutchfield as the enigmatic information[31]) could provide a measure of the overall balance between redundancy and synergy in multivariate systems: if TC($\mathbf{X}$) > DTC($\mathbf{X}$), then the global constraints on the system dominate and force a redundant dynamic, while if TC($\mathbf{X}$) < DTC($\mathbf{X}$) the system is dominated by information that is both shared, but not redundant. Rosas et al., rechristen this measure the organizational information:

$$\Omega(\mathbf{X}) = TC(\mathbf{X}) - \text{DTC}(\mathbf{X}) \qquad (8)$$

In the specific case of three variables, $\Omega(X_1, X_2, X_3)$ is equivalent to the co-information[24], which Williams and Beer showed is itself equivalent to the redundancy minus the synergy[51]. This is in keeping with the intuition that positive O-information implies a redundancy-dominated structure and a negative O-information implies a synergy-dominated structure, although the direct link between $\Omega$ and the co-information is only direct for three variables and the measures are not identical for larger sets.

While O-information has been applied in a variety of contexts (such as to questions about the aging brain[22], information flow in neuronal circuits[52], and music composition[16]), there remains considerable uncertainty around how synergy should be intuitively understood. To help elucidate the answer, we relate O-information to the original measure of integration/segregation balance proposed by Tononi, Sporns, and Edelman: the TSE complexity[9] and show that a geometric interpretation of the O-information exists that brings with it a novel perspective on redundancy and synergy.

The TSE-complexity admits two formulations:

$$\text{TSE}(\mathbf{X}) = \sum_{i=1}^{\lfloor N/2 \rfloor} \mathbb{E}[I(\mathbf{X}^{\gamma}; \mathbf{X}^{-\gamma})]_{|\gamma| = i} \qquad (9)$$

$$= \sum_{i=1}^{N} \left( \frac{i}{N} \text{TC}(\mathbf{X}) - \mathbb{E}[\text{TC}(\mathbf{X}^{\gamma})] \right)_{|\gamma| = i} \qquad (10)$$

The first (Eq. (9)) defines the TSE complexity as the average mutual information between the pairs of every possible bipartition of the system $\mathbf{X}$. For every integer $i$ between 1 and $\lfloor n/2 \rfloor$, we compute all possible subsets of $\mathbf{X}$ with $i$ elements (notated by $\mathbf{X}^{\gamma}$) and compute the mutual information between that set and it's complement ($\mathbf{X}^{-\gamma}$). The second equation (Eq. (10)) provides an alternative interpretation: the TSE complexity quantifies the difference, at every scale, between the expected integration of the scale if the system were fully integrated, and the actual integration of that scale (calculated as the average total correlation of every subset of size $k$). In this interpretation, the TSE complexity is highest when the smallest scales are relatively dis-integrated, but the macro-scales are relatively more integrated. This balance of integration and segregation is emblematic of TSE complexity. For a visualization of the TSE complexity calculation as the difference between the expected and empirical values, see Fig. 1.

Computing the full TSE complexity itself requires analyzing every possible subsystem (or bipartition) of $\mathbf{X}$: an insurmountable task for all but the smallest networks, as the combinatorics grow super-exponentially. A useful approximation is to look only at the second-to-top layer of the full TSE complexity summation, which

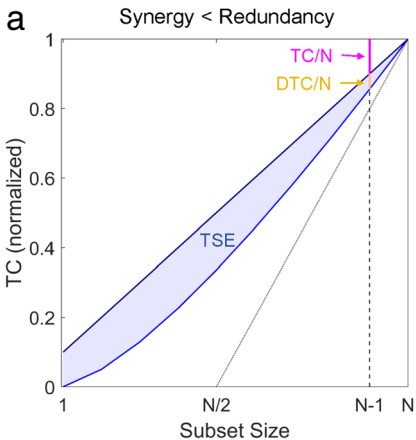
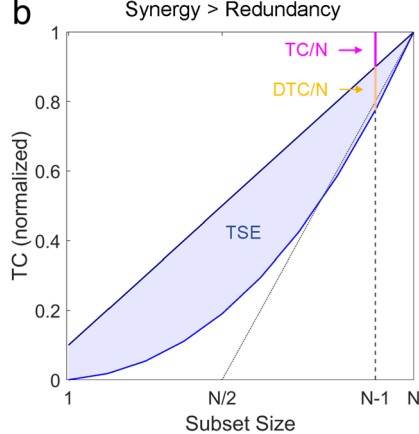

**Fig. 1 Understanding O-information in the context of the TSE complexity.** The TSE complexity provides a system-wide summary statistic of how integrated the system is at every scale. The O-information can be understood as measuring how sensitive the global integration is to the removal of single elements (on average). **a** The panel shows a TSE curve for a low-synergy system: $TC(\mathbf{X})/N > DTC(\mathbf{X})/N$, so $\Omega(\mathbf{X}) > 0$. The erasure of any single element, on average, does not change the overall integration of the remaining ($N-1$) elements much more than would be expected in the null case. **b** The panel shows the case where $TC(\mathbf{X})/N < DTC(\mathbf{X})/N$, so $\Omega(\mathbf{X}) < 0$ and the system is synergy dominated. Intuitively, this can be understood by recognizing that, on average, the removal of any of the $N$ elements causes a large decrease in the integration among the remaining ($N-1$) elements.

only requires finding the average total correlation for the $N$ sets $\mathbf{X}^{-i}$ (where $\mathbf{X}^{-i}$ is every $X \in \mathbf{X}$ excluding $X_i$). We refer to this measure as the description complexity of $\mathbf{X}$[10,53]. Formally:

$$C(\mathbf{X}) := TC(\mathbf{X}) - \frac{TC(\mathbf{X})}{N} - \mathbb{E}[TC(\mathbf{X}^{-i})] \quad (11)$$

The definition of $C(\mathbf{X})$ can be understood as the successive pruning of information: the first term, $TC(\mathbf{X})$, is the total integration of $\mathbf{X}$. The second term, $-TC(\mathbf{X})/N$, is the expected decrease in integration associated with a single element (on average). Finally, $-\mathbb{E}[TC(\mathbf{X}^{-i})]$, is the actual decrease in integrated associated with removing every element on its own. $C$, then, computes the difference between the expected decrease in integration associated with removing a single node and the actual decrease. $C$ has several obvious conceptual parallels with the DTC and there is indeed an analytic relationship between DTC and $C$ (for proof, see Supplementary Note 1):

$$DTC(\mathbf{X}) = N \times C(\mathbf{X}) \quad (12)$$

This result was independently derived in ref. [54]. The relationship between DTC and $C$ allows us to rewrite the O-information purely in terms of total correlations:

$$\Omega(\mathbf{X}) = TC(\mathbf{X}) - N \times C(\mathbf{X}) \quad (13)$$

$$= (2 - N)TC(\mathbf{X}) + \sum_{i=1}^{N} TC(\mathbf{X}^{-i}) \quad (14)$$

This allows us to re-conceptualize redundancy- and synergy-dominance in terms of just redundancy: synergistic information is information that is redundantly present in large ensembles of elements considered jointly but not in any subset of those ensembles. This is conceptually very similar to the definition of synergy provided by the partial information decomposition[27], which defines synergy in terms of redundant information shared by higher-order collections of elements. We can also propose a geometric interpretation of the sign of the O-information: based on Eqs. (8) and (13), we can see that $\Omega(\mathbf{X}) < 0 \Leftrightarrow TC/N < C$ and $\Omega(\mathbf{X}) > 0 \Leftrightarrow TC/N > C$. This means that a system $\mathbf{X}$ is synergy-dominated if the removal of a single element (on average) decreases the integration of the remaining $N-1$ elements more than would be expected in the null case of a totally integrated system. The two possible cases (redundancy-dominated, with

$\Omega > 0$ and synergy-dominated, with $\Omega < 0$) are visualized and discussed in the context of the TSE complexity in Fig. 1.

The framing of the O-information in terms of the change to integration after removal of individual elements also has conceptual links to the so-called gradients of O-information[55]. Scagliarini et al., explore how individual elements can contribute redundantly or synergistically to the O-information, defining the gradient as the difference between the O-information of an ensemble $\mathbf{X}$ and the O-information when single elements $X_i$ are excluded. While a detailed analytic exploration of the link is beyond the scope of this paper, the property of gradients yield valuable insights into the structure of higher-order dependencies in complex systems.

Another heuristic approximation of the TSE complexity is the sum of the total correlation and dual total correlation. Following the notation from Rosas et al.:

$$\Sigma(\mathbf{X}) = TC(\mathbf{X}) + DTC(\mathbf{X}) \quad (15)$$

James et al. previously termed this measure the exogenous information and described it as a very mutual information: quantifying all of the shared dependencies between each single variable and every other subset of the system:

$$\Sigma(\mathbf{X}) = \sum_{i=1}^{N} I(X_i; \mathbf{X}^{-i}) \quad (16)$$

Given the obvious similarity to Eq. (9), Rosas et al., hypothesized that $\Sigma(\mathbf{X}) \propto TSE(\mathbf{X})$, which was verified to hold in simple simulations with small $N$[24]. By leveraging the Gaussian assumptions here, we can empirically estimate the correlation between TSE and exogenous information and assess how well the relationship holds as $N$ gets large. Figure 2 confirms the strong correlations between TSE complexity with both TC + DTC and DTC alone. These correlations hold over a range of subset sizes, from three to fifteen elements.

**fMRI results.** We set out to identify subsystems (subsets of dynamically interacting elements) that express negative O-information (synergy) in the human brain. Leveraging Gaussian assumptions[42] (see Methods), multivariate information theoretic measures can be estimated from covariance (correlation) matrices expressing empirically recorded functional connectivity (FC). We computed long-time averages of FC derived

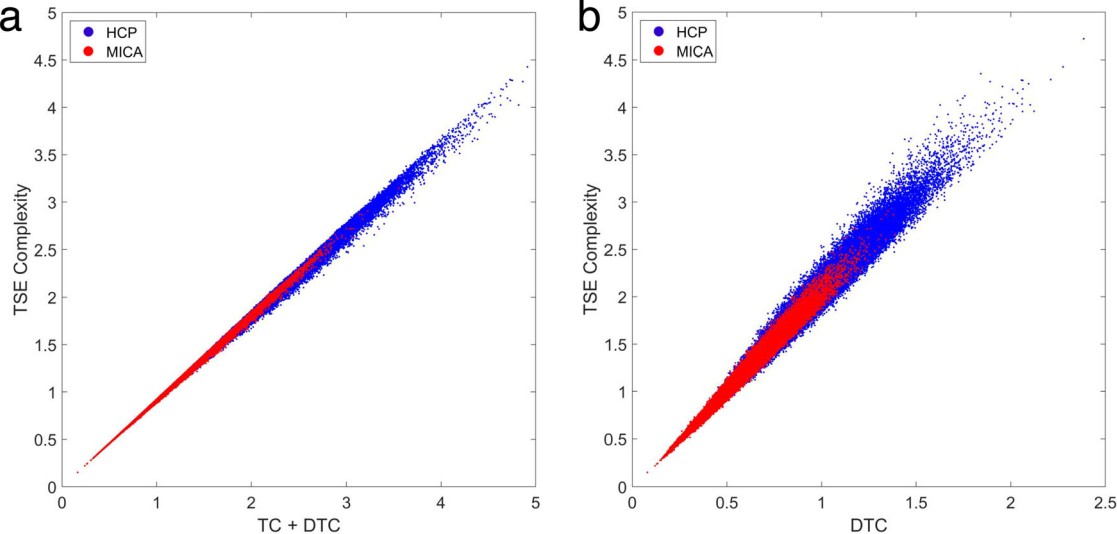

**Fig. 2 Approximating TSE complexity with total correlation and dual total correlation.** Data are from 100,000 randomly selected subsets of 10 nodes (blue: HCP data; red: MICA data), with TSE complexity computed exactly, by sampling all subsystems. **a** Sum of TC + DTC versus TSE complexity; HCP: $R = 0.998, p = 0$; MICA: $R = 0.999, p = 0$. **b** DTC versus TSE; HCP: $R = 0.982, p = 0$; MICA: $R = 0.992, p = 0$.

from two normative samples of human resting-state fMRI, the Human Connectome Project (main data set[56]) and an open-source multimodal MRI dataset for Microstructure-Informed Connectomics (MICA-MICs; replication data set;[57]). For both data sets we computed a single FC matrix (HCP: 95 participants, 4 runs each; MICA-MICs: 50 participants, 1 run each). Both FC covered the entire cerebral cortex parcellated into a common set of 200 nodes[58] and node time series were derived from BOLD signals after performing global signal regression, which removes signal components that are common to all nodes in the system, i.e., globally redundant (Supplementary Fig. 1)[59]. For a brief discussion of global signal regression in this context, see Supplementary Fig. 6).

Computing O-Information on the full-size 200-node FC matrix results in positive quantities for both data sets (HCP: $\Omega = 79.16$ nats; MICA: $\Omega = 46.69$ nats), indicating that the full structure is redundancy-dominated, which might potentially obscure the presence of higher-order, synergistic correlations. We asked if smaller subsets of nodes were present within the full-size FC that generated synergy, or negative O-information. Random sampling of small subsets (between 3 and 16 nodes) indeed yields abundant subsets that express negative O-information (Fig. 3a). Their relative abundance declines rapidly with growing subset size, reflecting the increasing dominance of redundant information and exhaustive capture of unique information. While synergistic subsets account for rapidly diminishing fractions of all subsets, their total number can be non-negligible (10-node subsets: 0.41 percent and $9.23 \times 10^{13}$, respectively). In a large random sample of 10-node subsets, the O-information is positively correlated with TSE complexity (Fig. 3b; $\rho = 0.642, p = 0$; HCP data). Focusing on a separate random sample of 5000 10-node subsets with negative O-information, we asked if the frequency with which pairs of nodes participate in such subsets is related to their pairwise FC. Indeed, the absolute pairwise FC is strongly negatively correlated with the frequency of participation in synergistic subsets ($\rho = -0.504, p = 0$, HCP; $\rho = -0.485, p = 0$, MICA, HCP; Fig. 3c). This indicates that strongly positive or negative FC between two nodes makes their joint inclusion in a synergistic subset unlikely, while node pairs with low FC magnitude could either be truly disintegrated, or participating in a highly synergistic subsystem.

Participation of nodes in randomly sampled synergistic subsets varies systematically across the cortex. Over a large random sample of 100,000 10-node subsets, all nodes participate at least once, with several nodes participating in more than 10,000 distinct synergistic subsets. Hence, the complete repertoire of co-expressed synergistic subsets covers the entire cortex, with some overlap between subsets, centered on high-participation nodes that form "focal points" or clusters (Fig. 4a). Projecting the participation of individual nodes (brain regions) onto the cortical surface shows significant consistency between HCP (Fig. 4b) and MICA data (Supplementary Fig. 4) (the two maps are correlated with $\rho = 0.579, p = 2.5 \times 10^{-19}$, between the two data sets). Functional systems[60] distribute unevenly as well, with highest frequencies of participation found in the frontoparietal (FP) system, for synergistic subsets of 10 nodes (HCP: Fig. 4c; MICA: Supplementary Fig. 2). For larger subset sizes, participation of limbic (LIM) regions dominates over FP regions.

Combinatorics prevent exhaustive exploration of subsets of even modest sizes, and the random sampling strategy employed so far is likely to miss subsets that express maximal synergy. To identify subsets with maximally negative O-information (maximal synergy), we used an optimization algorithm based on simulated annealing (references and details are contained in the Methods section). Multiple runs of the algorithm yielded consistent and highly similar outcomes (Supplementary Fig. 3), indicating convergence of the optimization while again high-lighting the existence of a large reservoir of non-identical (degenerate) subsets, all expressing highly negative O-information. Deploying this algorithm while varying subset sizes between 3 and 30 nodes, we identified large numbers of subsets that express highly negative O-information, for subset sizes 3-24 nodes (HCP; Fig. 5a) and 3–27 nodes (MICA; Supplementary Fig. 4). No synergistic subsets are found for subset sizes greater than 27 nodes, as redundancy starts to overwhelm the unique informational contributions of individual nodes at larger subset sizes.

To validate that our optimization algorithm was observing truly synergistic ensembles, we tested each optimized subsystem against a null (see Materials and Methods E). Since the ensemble size $k$ is fixed by the optimization algorithm, it is possible that the apparent synergy of that ensemble is actually due to some subset

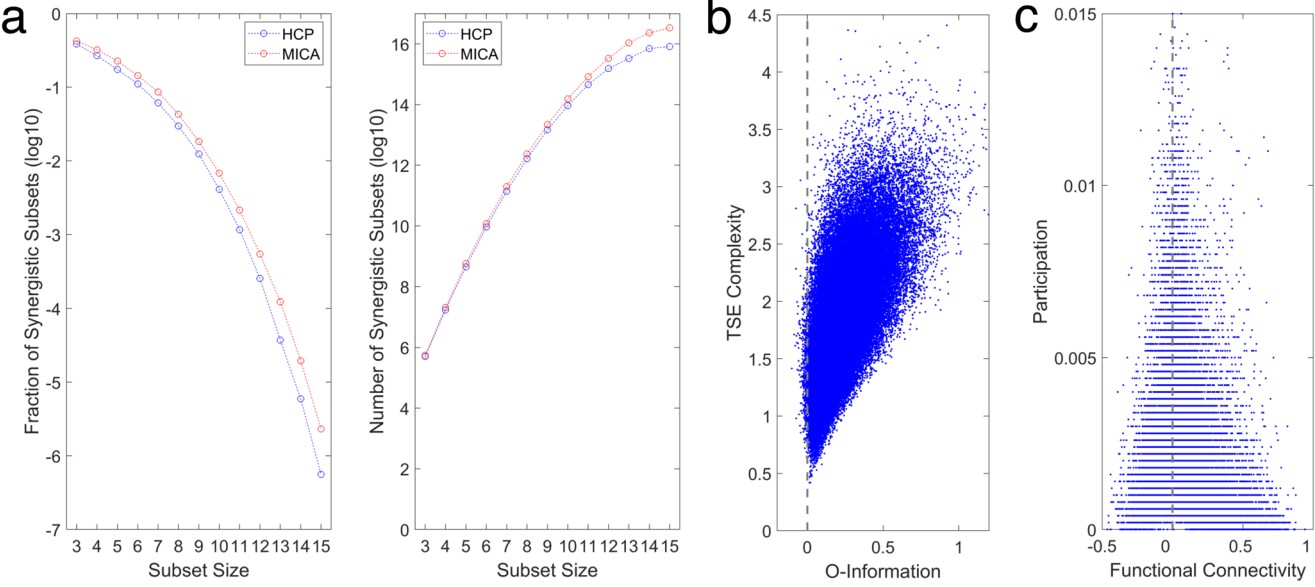

**Fig. 3 Information measures computed from randomly sampled 10-node subsets. a** Fraction (left) and number (right) of subsets with negative O-information, obtained by randomly sampling subsets from the HCP (blue) and MICA (red) FC matrix. Fraction and number estimated from samples of 5000 (3–12 nodes) or 200 (13–15 nodes) subsets with negative O-information. As the size of the subset grows, the fraction expressing an overall synergy-dominated structure (negative O-information) drops precipitously, while their absolute number continues to climb due to combinatorics. **b** The relation between O-information and TSE complexity in 100,000 randomly sampled 10-node subsystems (HCP data). While very few randomly sampled sets have negative O-information (see panel **a**), TSE complexity generally increases with the strength of the dependencies visible to the O-information ($R = 0.642, p = 0$). **c** The participation quantifies, for each node pair, how often they are encountered as part of a subset with negative o-information (10 nodes, 5000 random samples, HCP data), The plot shows the relation of the participation against the FC, with each data point representing one of the 19,900 unique node pairs. Node pairs with strong mutual FC (positive or negative) are rarely encountered as part of the same synergistic subset, while node pairs that are more frequently encountered tend to show weak FC. Spearman's rho between absolute FC and participation: $\rho = -0.504, p = 0$.

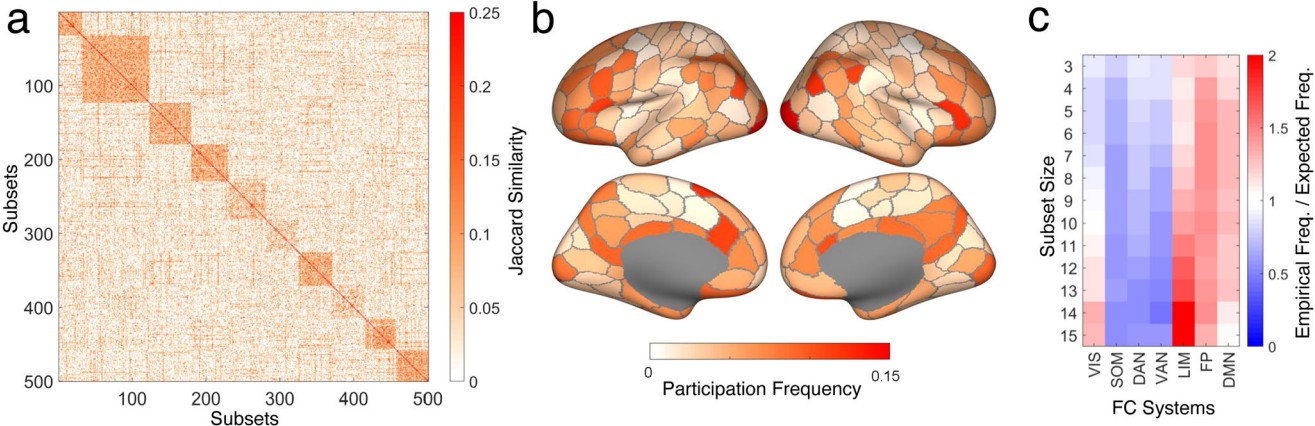

**Fig. 4 Topography and functional specialization of randomly sampled synergistic subsets in the brain.** Data in panels **a** and **b** was derived from a random sample of 100,000 synergistic 10-node subsets (HCP data). **a** Drawing random sub-samples of 500 subsets, we computed their Jaccard similarity, capturing the number of nodes in common between each subset pair. The similarity matrix was clustered using the kmeans algorithm, iterating between 2 and 30 clusters, with 10,000 repetitions. Optimal cluster quality was determined using the 'silhouette' criterion on the resulting cluster assignments. Random samples consistently yielded around 9–11 optimal clusters, with one example (10 clusters) shown in this panel. A Jaccard similarity of 0.25 corresponds to two subsets having 4 out of 10 nodes in common. **b** Frequency of individual node participation across 100,000 synergistic subsets, displayed on a surface rendering of the cerebral cortex indicating the boundaries of the 200 nodes used for constructing the FC matrix. **c** Each of the 200 nodes is affiliated with one of 7 canonical functional systems[60]. Frequency of participation of individual nodes in synergistic subsets (negative O-information, subset size ranging from 3 to 15 nodes) is aggregated (averaged) for each functional system. The plot displays the ratio of empirical frequency over the expected frequency if nodes were selected by chance. A ratio > 1 or < 1 indicates that the system is over-represented or under-represented, respectively, in synergistic subsets. Sample sizes identical to those used in Fig. 4a.

of nodes within that ensemble (for example, a system of three synergistic elements and two independent elements will appear to be a synergistic system of $k = 5$, however, the real synergy is only in the three entangled elements). To ensure that all elements in

the ensemble contributed to the synergy, we only considered a set valid if it was impossible to remove any node without the O-information increasing (i.e., the contribution of each element was synergy-dominated). We found that, for small value of $k$, the

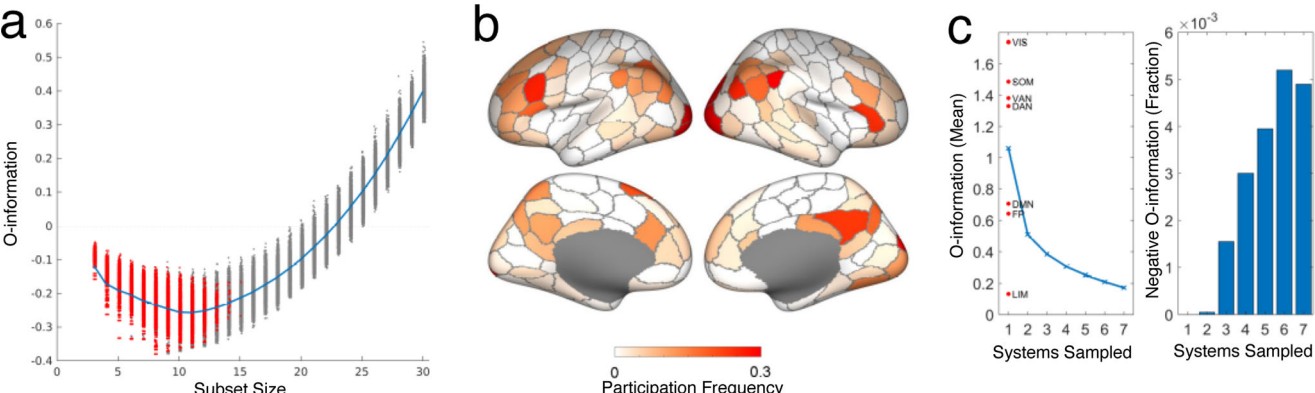

**Fig. 5 O-Information, brain topography, and functional specialization of optimally synergistic subsets identified by simulated annealing All panels show data from the HCP sample. a** Annealing was carried out 5000 times for each subset size. This plot shows O-information for each optimized subset (gray dots) and their mean (blue line). Note that annealing fails to converge onto any synergistic subsets for subsets containing more than 24 nodes. Optimally negative O-information is achieved for subsets between 8 and 12 nodes. For each subset of size k each node was removed individually and the O-information of the remaining $k-1$ nodes was calculated. If the O-information was lower than given by the original $k$ node subset, the contribution of that node was considered redundant and the synergy in the subset attributable to its $k-1$ node counterpart. For the vast majority of subsets smaller than 12 nodes, no nodes could be removed in a way that increased synergy, indicating that these subsets consisted of nodes with irreducible synergy (red in Fig. 5a). **b** Frequency of individual node participation across optimally synergistic 10-node subsets (4021 unique subsets out of 5,000 annealing runs), displayed on a surface rendering of the cerebral cortex (cf. Fig. 4b). **c** Mean O-information (left) and fraction of synergistic subsets (right) encountered in samples of 20,000 subsets that contained nodes belonging to between 1 and 7 canonical FC systems (HCP data). The mean O-information for samples obtained exclusively from each of the 7 FC systems is indicated (red dots).

vast majority of optimized ensembles were valid (≈99.08% for ensembles of size four, ≈92.92% for ensembles of size six, ≈84.14% for ensembles of size eight, and ≈64.04% for the maximally synergistic ensemble size of ten). For collections much larger than ten, the proportion of valid systems decreased rapidly: for ensembles of size fifteen, only ≈0.04% were valid, and there were no valid ensembles of size greater than fifteen, despite the fact that the simulated annealing algorithm returned a large number of results with $\Omega < 0$. This implies that, although these larger subsystems are synergy dominated, that synergy is restricted to a core set of components and not attributable to the whole.

Minimal O-information was achieved for subsets comprising ~10 nodes for both data sets. Mapping subsets of nodes expressing near minimal O-information onto a surface plot of the cerebral cortex reveals consistent topography. Figure 5b shows the frequency with which individual cortical parcels (nodes) were identified across 5000 runs of the optimization algorithm, yielding 4021 unique solutions (HCP Fig. 5b; 4166 unique solutions for MICA data, Supplementary Fig. 4b). Brain-wide nodal frequencies are significantly correlated across HCP and MICA data sets (Spearman's $\rho = 0.522$, $p = 2.2 \times 10^{-15}$). When mapping these nodal frequencies to seven canonical resting-state functional systems[60], we find that each of these seven systems contributes, but to different extent. In HCP data, for optimally synergistic 10-node subsets, the visual, frontoparietal and default mode networks are over-represented, while only the FP system appears over-represented in the MICA data (Supplementary Fig. 5).

The nature of negative O-information (synergy) requires that individual nodes make largely unique (non-redundant) contributions to the multivariate information metric. This suggests that nodes derived from different, informationally distinct (intrinsically redundant, but extrinsically non-redundant) functional communities might be favored as constituents of synergistic subsets. To test this hypothesis, we created sets of 20,000 randomly sampled subsets that were comprised of nodes derived from between 1 and all 7 canonical functional systems (HCP, Fig. 5c; MICA, Supplementary Fig. 4c). The mean O-information,

across all randomly chosen subsets, was found to be positive regardless of how many FC systems were included in the subsets. For samples derived from just 1 FC system, the O-information was most positive (i.e., subsets were most redundancy-dominated) for visual, somatomotor and attention systems, and they were least redundancy-dominated for default, frontoparietal and limbic systems. Importantly, the mean O-information decreased, and the fraction of synergstic subsets increased, as subsets were sampled from larger numbers of canonical systems. No subset derived from a single functional system was capable of expressing synergy. Subsets spanning 6 or 7 canonical systems were most likely to express synergy, as indexed by the fraction of negative O-information encountered in the sample. The finding supports the notion that dividing the brain into canonical functional systems prioritizes grouping nodes by redundant over synergistic information, hence missing a potentially important substrate for neural computation.

## Discussion

In this paper, we have shown how the O-information[24], a measure of higher-order interactions in multivariate data, can reveal synergistic ensembles of brain regions that are invisible to bivariate functional connectivity analyses. Our primary theoretical result is to provide a geometric interpretation of the O-information. The interpretation unifies multiple disparate measures of multivariate information into a single framework, built around the Tononi-Sporns-Edelman complexity[9]. By re-writing $\Omega(\mathbf{X})$ and $DTC(\mathbf{X})$ in terms of the total correlation between multiple subsets of $\mathbf{X}$, we find that synergy occurs when removing any single element causes the system to become less integrated, and crucially, more-so than would be expected if structure was uniformly distributed over $\mathbf{X}$. Said differently, synergy can be intuitively understood as that integration that is present in the whole but not smaller subsets (in this case, the $N$ subsets created by removing each $X_i$). In this sense, synergy captures how the whole can be greater than the sum of it's parts[61]. This intuition is conceptually similar to the formal definition of synergy from the partial information decomposition framework[27], which defines synergy as the information left over

when everything accessible in simpler combinations of sources has been accounted for. The exclusive use of total correlations also allows us to consider the O-information purely in terms of Kullback-Leibler divergences from independent to joint probability distributions (Eq. (5)). This shows us that all of these measures can be understood in the context of inferences about structure (relative to a disintegrated prior). In the context of synergy, the extra information in the joint state is information about something: specifically about the relative likelihood of a configuration with respect to the maximum entropy case.

Applied to two separate fMRI brain data sets we find that synergistic subsets of brain regions are ubiquitous and abundant, spanning scales between 3 and 25 regions and extending over the entire cerebral cortex. While redundant interactions dominate functional connectivity at larger subset sizes, the application of multivariate information measures demonstrates a previously hidden repertoire of synergistic ensembles, each integrating diverse and distinct sources of information. Recent work by Luppi et al.,[21] proposed a synergistic core to the human brain where complex processing occurs. While we found that there is significant over-representation of specific regions (including portions reported by Luppi et al., such as prefrontal cortex, occipital pole, the precuneus, and cingulate regions), synergy-dominated subsystems could include any region of cortex, although some regions contribute more reliably than others. This suggests that synergy is a widespread property of multivariate information emerging from resting-state brain activity. While there is discrepancy between our results and those of Luppi et al., this difference is likely a reflection of the different analytical pipelines, rather than a true conflict. Their approach was based on decomposing the temporal mutual information, which considers dependencies between past and future, while our approach does not. Luppi et al., also only considered pairs of regions co-evolving together, while we considered larger ensembles. Our approach brings into view synergies of a much higher order than would be possible in the approach by Luppi et al. Finally, the prior analysis is based on a generalization of the partial information decomposition and requires choosing one of several redundancy functions. It is unknown whether the reported results would hold for all plausible redundant information functions or not. Consequently, different results likely reflect different kinds of synergies (temporal, pairwise vs. instantaneous, higher-order) that can co-exist in the brain.

Information theoretic measures are not the only approaches to higher-order structure in brain activity. Recently, Santoro et al.,[62] proposed a metric they term hyper-coherence, which describes higher-order co-fluctuations between sets of three and four regions. Based on the edge time series framework[63–65], the hyper-coherence defines a higher-order activity in terms of simplicial complexes of coherent activations. When applied to brain data, Santoro et al., found that hyper-coherence was relatively lower in systems that we found to be relatively higher in synergy (frontal and default mode regions), and relatively higher in regions we found to have high average O-information (somato-motor regions). We conjecture that the hyper-coherence framework is probably preferentially sensitive to redundancies rather than synergies. This would be consistent with recent results from Varley et al., who found that pairwise co-fluctuations were positively correlated with redundant information and anti-correlated with synergistic information[66]. An avenue of future research may be to attempt to apply the filtering approach Santoro et al., use to the O-information or other multivariate information measures.

Interestingly, the randomly sampled ensembles that were most likely to be synergy dominated were those that involved nodes that spanned multiple canonical subsystems, while sets of regions all within one system were strongly dominated by redundancy (Fig. 5c). This would be consistent with the hypothesis that functional connectivity, when viewed entirely as bivariate interaction, is largely sensitive to redundant, but insensitive to synergistic, dependencies between brain regions. Consequently, the functional connectivity matrix is not a complete map of the statistical structure in a dataset, but only of dependencies characterized by redundancy. This is consistent with findings from Ince[67], Finn and Lizier[68], and Varley et al.[66], who argued that bivariate correlations are intrinsically redundancy-dominated. Higher-order synergies represent, in a sense, a kind of shadow structure and consequently are missed by network-focused approaches that omit higher-order interactions. This hypothesis finds some support in ref. [21], who found that the distribution of synergies was anticorrelated with the functional connectivity network structure, while the distribution of redundancies was positively correlated.

Given the novelty of tools like the O-information, the significance of these synergistic dependencies remains almost entirely unknown, although the small number of studies to date suggest intriguing patterns. One study found alterations to the redundancy/synergy bias across the human lifespan[22], while other studies have suggested that loss of consciousness induced by propofol is associated with decreased synergistic dynamics[20]. Future avenues of work include deeper analyses of how higher-order dynamics change between rest and task conditions, in cases of psychopathology or brain injury, and non-human animals. We should note that, in the context of the O-information, synergy is not necessarily a causal measure: in related contexts, synergy has been discussed as a measure of computation in neural circuits by Sherrill et al., Newman et al., and others[13–15,17,26,69], although it remains unexplored how exactly these two approaches relate to each-other. The O-information is an atemporal measure, sensitive to instantaneous, higher-order correlation structures, but with no notion of dynamics or a flow from past to future. In contrast, the work by Sherill et al., is done in the context of information dynamics[25] and considers how the past informs on the future. Future research may explore how a synergistic correlation structure might facilitate computations within the system over time.

In addition to the insights into synergy specifically, the results presented here also have implications for researchers interested in multivariate information theoretic analyses. For example, the TSE complexity has long been an object of theoretical interest[54], but the intractable combinatorics have limited its applicability in empirical data (although its use is not unheard of[70]). The finding that the exogenous information $\Sigma(\mathbf{X}) \propto \mathrm{TSE}(\mathbf{X})$ for reasonably large $N$ (first reported in ref. [24]), even more so than the original heuristic $C$, opens the door to applications in experimental neuroscience. The nature of this correlation merits further study as well. One outstanding question is how redundancies and synergies in the data differentially influence the relationship between $\Sigma(\mathbf{X})$ and $\mathrm{TSE}(\mathbf{X})$. Unlike the O-information, the S-information does not obviously link to redundancies or synergies, and so how these kinds of integration impact the relationship to TSE remains unknown. Future work developing generative models with precisely controllable distributions of redundancies and synergies may shed light on this question.

In a broader scientific context, our work contributes to the increasing interest in higher-order interactions, beyond the standard, pairwise network model[8,71]. The information-theoretic approach (such as the work reported here, as well as in refs. [16,20,21,52,69,72]) is based largely on a statistical inference, while alternative frameworks based on simplicial complexes, algebraic topology, and hypergraphs has been developed largely in parallel[7,41,73–76]. How these different mathematical

frameworks relate to each other remains an open question, and the potential for a more unified approach to understanding higher-order interactions both in terms of topology and statistical inferences is an alluring promise.

The optimization of maximally synergistic subsets via simulated annealing can be thought of as an attempt to find a maximally efficient, dimensionally reduced representation of a potentially large data set: when modeling a system, it is generally desirable to capture as many statistical dependencies as possible with the fewest required degrees of freedom. By finding a representation that incorporates synergies while simultaneously pruning redundant information that would be double counted, we can attempt to build the most computationally efficient model of a system under study[77,78]. While dimensionality reduction and feature selection algorithms are widespread in many computational sciences, a rigorous treatment of the ways that synergistic and redundant information can inform the analysis of brain dynamics and functional networks remains a space of active development (for an example, see refs. [78,79]).

The O-information scales far more gracefully than related measures of synergistic information (such as the partial information decomposition, which is practically impossible to apply to systems larger than 5 elements[28]). However, the combinatorics associated with assessing every possible subsystem becomes intractable as the system size grows, an issue first noted for the TSE complexity. In standard functional and effective network research, it is common to compute all pairwise interactions (which only grows with $N^2$), and then filter out spurious edges as needed[79]. While this may be possible for very small subsystems, it is intractable for larger ones. If one can pre-select a set of elements, then the computation of O-information is trivial up to hundreds of items. However, the requirement to select subsets of interest can itself be computationally intensive and time-consuming. Consequently heuristic measures such as optimization, random sampling, or pre-filtering subsystems to exclude collections of elements will be required.

Since the O-information is a measure of relative redundancy/synergy dominance, in highly redundant data, synergistic structures may not be strong enough to dominate the signal, resulting in a positive (redundancy-dominated) O-information. By adding increasing amounts of low-frequency redundancy to the BOLD data, and re-running the optimizations, we found that the maximally synergistic subsets extracted from the uncontaminated data became impossible to retrieve (see Supplementary Fig. 6). Those synergies still existed, they were merely swamped by redundancy and made invisible. Adding global signal back in this way provides a new insight into a commonly used step in fMRI image pre-processing: global signal regression (GSR)[80]. We argue that GSR can be understood as scrubbing global redundancies from the data, and in doing so may reveal previously buried synergies that would not have been accessible in the original, unprocessed data.

One limitation of this study is that it is hard to disambiguate between information that reflects computation in neural tissue, versus what is attributable to the vascular physiology of the BOLD signal. Recent work by Colenbier has shown that there are synergistic interactions between the global signal, blood arrival times, and functional connectivity structure[59]. Since the pairwise covariance forms the foundation of the multivariate Gaussian entropy estimator, it is likely that the same confounds influence the estimates of entropy and mutual information. Future work replicating these results using electrophysiological recordings such as M/EEG should help untangle this issue. Another limitation is that it operates on static distributions: every frame is assumed to have been drawn from an unchanging multivariate Gaussian distribution, with no memory or dynamics from moment to moment. This is a standard assumption in functional connectivity analyses, although there is growing interest in the limitations this assumption produces and the need for analyses that explicitly account for dynamics[81]. The field of information dynamics provides a number of relevant analyses[82,83], and there is already interest in higher-order dynamics in the brain: in addition to the aforementioned work by Luppi et al., recent work by Faes et al. proposed a derivative of the O-information for rhythmic processes (the O-information rate, or OIR)[84]. The OIR has been used the describe brain-heart interaction dynamics and opens up the frequency domain to higher-order, informational analysis in addition to the time domain. Similarly, an application of the O-information to the dynamic measure of transfer entropy has been proposed and applied to optimizing ensembles of maximally synergistic or redundant neurons[52]. Both of these measures could be incorporated into a pipeline line the one described here and may shed light on the similarities (and differences) between dynamic and static analyses. Despite its limitations, however, we are confident that the classic, static O-information likely contains a wealth of as-yet unexplored structure and will continue to provide insights into brain structure and function.

In this article, we demonstrate how an information-theoretic measure of multivariate interactions (the O-information or synergy) can be used to uncover higher-order interactions in the human brain dynamics. We analytically show that the O-information can be related to an older measure of systemic complexity, the TSE complexity, and from this derive a novel geometric interpretation of redundancy- and synergy-dominated systems. With a combination of random sampling and optimization, we show that a large number of subsystems displaying synergistic dynamics exist in the human brain and that these systems form a highly distributed shadow structure that is entirely overlooked in standard, bivariate functional connectivity models. We conclude that the space of higher-order interactions in the human brain represents a large, and under-explored area of study with a rich potential for new discoveries and experimental work.

## Methods

**Gaussian information theory**. In this paper, we focus on higher-order information sharing in fMRI BOLD signals. Since BOLD data is continuous (rather than discrete), to quantify the entropy of a continuous signal, we use a generalization of the the classic, discrete Shannon entropy (Eq. (1)): the differential entropy:

$$H(X) = \int_{x \in \mathcal{X}} P(x) \log P(x) dx \qquad (17)$$

Computing the differential entropy from empirical data is generally difficult, as it requires estimating $P(x)$. However, if one is willing to make assumptions of multivariate normality, closed-form estimators of the Gaussian joint entropy can be leveraged.

Prior work has established that BOLD data is well-modeled by multivariate Gaussian distributions[85,86] and that more complex and highly parameterized models provide little additional benefit[87]. While information theory was originally formalized in the context of discrete random variables, in the specific case of Gaussian random variables, closed-form estimators exist for almost all the standard information measures (for an accessible review, see[82] supplementary material). For a univariate, Gaussian random variable $X \sim \mathcal{N}(\mu, \sigma)$, the entropy (given in nats) is defined as:

$$H^{\mathcal{N}}(X) = \frac{\ln(2\pi e \sigma^2)}{2} \qquad (18)$$

For a multivariate Gaussian random variable $\mathbf{X} = \{X_1, X_2, \ldots X_N\}$, the joint entropy is given by:

$$H^{\mathcal{N}}(\mathbf{X}) = \frac{\ln[(2\pi e)^N |\Sigma|]}{2} \qquad (19)$$

where $|\Sigma|$ refers to the determinant of the covariance matrix of $\mathbf{X}$. The bivariate mutual information (nats) between $X_1$ and $X_2$ is:

$$I^{\mathcal{N}}(X_1; X_2) = \frac{-\ln(1 - \rho^2)}{2} \qquad (20)$$

where $\rho$ is the Pearson correlation coefficient between $X_1$ and $X_2$. Note that, since the mutual information is a function of $\rho$ for Gaussian variables, this special case of mutual information is **not** generally sensitive to non-linear relationships in the data in the way that non-parametric estimators are. Finally, the Gaussian estimator for total correlation is:

$$TC^{\mathcal{N}}(\mathbf{X}) = \frac{-\ln(|\Sigma|)}{2} \tag{21}$$

From these, it is possible to calculate all of the measures described above (dual total correlation, description complexity, O-information, and TSE complexity) for multivariate Gaussian variables. While the assumption of linearity that comes with a parametric Gaussian model can be limiting, the standard technique for assessing functional connectivity (the Pearson correlation coefficient) makes identical assumptions, so our work is consistent with assumptions made when applying standard approaches to FC analysis.

Building an intuitive understanding of synergy in the context of linear systems is difficult, since a multivariate Gaussian is defined in terms of pairwise covariances. Barrett showed that higher-order synergies can exist in purely Gaussian systems and that redundancy is related to the mutual information[88], so even in linear systems, beyond-pairwise dependencies can exist. This can be partly understood by recognizing that the multivariate Gaussian is the maximum entropy distribution subject to the constraints of pairwise covariance[42]. So, while pairwise linear relationships are enough to uniquely specify the distribution, they do not rule out the possibility that beyond-pairwise relationships exist. They do, however, fix the structure of those higher-order dependencies.

**Datasets**. Two independent fMRI resting state data sets were employed in the empirical analyses, one derived from the Human Connectome Project (HCP data[56]) and the other from a recently published open-source repository (MICA[57]). The HCP data, derived from a set of 100 unrelated subjects, have been used in several previous studies (for more detailed description see ref.[89]). All participants provided informed consent, and the Washington University Institutional Review Board approved all of the study protocols and procedures. A Siemens 3T Connectom Skyra equipped with a 32-channel head coil was used to collect data. Resting-state functional MRI (rs-fMRI) data was acquired during four scans on two separate days. This was done with a gradient-echo echo-planar imaging (EPI) sequence (scan duration: 14:33 min; eyes open). Acquisition parameters of TR = 720 ms, TE = 33.1 ms, 52° flip angle, isotropic voxel resolution = 2 mm, with a multiband factor of 8 were used for data collection. A parcellation scheme covering the cerebral cortex developed in ref.[58] was used to map functional data to 200 regions. This parcellation can also be aligned to the canonical resting state networks found in ref.[60].

Of the 100 unrelated subjects considered in the original dataset, 95 were retained for inclusion in empirical analysis in this study. Exclusion criteria were established before the present study was conducted. They included the mean and mean absolute deviation of the relative root mean square (RMS) motion across either four resting-state MRI scans or one diffusion MRI scan, resulting in four summary motion measures. Subjects that exceeded 1.5 times the interquartile range (in the adverse direction) of the measurement distribution in two or more of these measures were excluded. Following these criteria, four subjects were excluded. Due to a software error during diffusion MRI processing, one additional subject was excluded. The remaining 95 subjects were 56% female, had a mean age of 29.29 ± 3.66, and an age range of 22 to 36.

The MICA dataset includes 50 unrelated subjects, who also provided written informed consent. The study was approved by the Ethics Committee of the Montreal Neurological Institute and Hospital. Resting state data were collected in a single scan session using a 3T Siemens Magnetom Prisma-Fit with a 64-channel head coil. Resting state scans lasted for 7 minutes during which participants were instructed to look at a fixation cross. Imaging was completed with an EPI sequence, and acquisition parameters of TR = 600 ms, TE = 48 ms, 52° flip angle, isotropic voxel resolution = 3 mm, and multiband factor 6. The parcellation used in this dataset was the same as the one used for the HCP data (described above).

**Preprocessing**. Minimal preprocessing of the HCP rs-fMRI data followed these steps[90]: (1) distortion, susceptibility, and motion correction; (2) registration to subjects' respective T1-weighted data; (3) bias and intensity normalization; (4) projection onto the 32k_fs_LR mesh; and (5) alignment to common space with a multimodal surface registration[91]. The preprocessing steps described produced an ICA+FIX time series in the CIFTI grayordinate coordinate system. Two additional preprocessing steps were performed: (6) global signal regression and 7) detrending and band pass filtering (0.008 to 0.08 Hz)[92]. After confound regression and filtering, the first and last 50 frames of the time series were discarded, resulting in a final scan length of 13.2 min (1100 frames).

Preprocessing of the MICA dataset was performed as described in ref.[57] for resting state data. Briefly, the data was passed through the Micapipe[93] processing pipeline, which includes motion and distortion correction, as well as FSL's ICA FIX tool trained with an in-house classifier. Time series were projected to each subject's FreeSurfer surface, where nodes were also defined. Further details about the processing pipeline can be found in ref.[93]. The data was global signal regressed in addition to the other preprocessing steps described in this pipeline.

For calculating the covariance matrix used in computing O-information, total correlation and dual total correlation, the functional data from all scans and all subjects were combined to create a single COV or FC matrix. Aggregation was carried out by appending the nodal time series across all subjects and runs and then calculating a single Pearson correlation for each node pair. An alternative approach (taking the mean over the single-run, single-subject COV/FC matrices) yielded virtually identical results. Following preprocessing and using the common 200-node parcellation of cerebral cortex, the mean COV/FC matrices for the HCP and MICA data sets were highly correlated ($R = 0.851$, $p = 0$).

**Random sampling and optimization**. Subsets of regions were selected from the full-size (200 nodes/regions) FC matrices in two ways, by random sampling and by search through optimization. Random sampling is simple to implement but because of the vast repertoire of potential subsets ($\binom{N}{k}$) it cannot fully disclose the extent of variations in informational measures present in the data. Instead, search under an objective function (optimization) can guide exploration to specific sub-spaces enriched in subsets with distinct informational signatures.

To perform optimizations we implemented a variant of simulated annealing[94]. As objectives we chose multivariate informational measures such as the O-information (OI), total correlation (TC), and dual total correlation (DTC), which could be maximized or minimized. Each run of the simulated annealing algorithm was carried out in one FC matrix and for one subset size. We carried out 5000 runs, with subset sizes ranging from 3 to 30 nodes. A random selection of nodes was chosen according to the given subset size to initiate each run. The corresponding covariance matrix was extracted from the full COV/FC and used to compute the information theoretic metric of interest. The composition of the subset was then varied and variations were selected under the objective function. Annealing operates by selecting variations stochastically, depending on a temperature parameter that determines the amount of noise permitted in the selection process. Initially, the temperature is high, resulting in the somewhat random exploration of the landscape. As the temperature is lowered, the optimization becomes more deterministic, focusing more and more on local gradient descent. For each run the algorithm proceeded for a maximum of 10,000 steps. At each step, a new set of nodes was generated by randomly replacing nodes, with the number determined by a normal distribution (frequencies of 1, 2, and 3 element flips were 0.68, 0.27 and 0.04, respectively). A new covariance matrix was computed for the new set of nodes and the objective function was calculated for that set. The set was retained if its cost was lower than the current set or if a random number drawn from the uniform distribution between 0 and 1 was less than $\exp(-((C_n - C)/T_c))$, where $C_n$ is the cost of the new set of nodes, $C_L$ is the cost of the current set of nodes and $T_c$ is the current temperature. At each step, the current temperature decays to a fraction of the initial temperature, as a function of the number of steps completed:

$$T_c(h) = T_0 \times (T_{\exp})^h \tag{22}$$

where $T_c$ is the current temperature, $T_0$ is the initial temperature (set to $T_0 = 1$), $T_{\exp}$ governs the steepness of the temperature gradient, and $h$ is the current iteration step. By decreasing the temperature at every step, the algorithm becomes progressively more deterministic.

**Null model**. Given some set $\mathbf{X}$ of $k$ nodes with $\Omega(\mathbf{X}) < 0$, it is possible that not every $X_i \in \mathbf{X}$ actually contributes to the synergy (for example, if there are some $X_i$ that are independent from every other node). However, this set may still be found as a solution for an optimally synergistic subset of $k$ nodes by the simulated annealing algorithm. To ensure that the synergy found in a given subset is the maximally synergistic set of nodes, each node in the subset was removed from the subset in turn by setting the Pearson correlation of that node with all other nodes to zero. After removal of a node, the O-information was recalculated. If removal of any node decreased the O-information of the subset, then the subset was considered reducible to the $k - 1$ subset, and was not included in further analyses.

**Statistics and reproducibility**. All statistics were computed using MATLAB 2020 and MATLAB 2021. The code for reproducing results is provided Supplementary Software 1. Covariance matrices were computed from z-score BOLD time series and squared off to ensure symmetry by averaging each matrix and its own transpose. Information-theoretic estimators were computed using the formulae given in Sec. Gaussian Information Theory (all code provided).

Random sampling of ensembles was done for ensembles of size 3–16, with 100,000 samples done for each size. Annealing was done using the provided code, with 5000 replications for each ensemble size. All correlations computed using Spearman's $\rho$.

**Reporting summary**. Further information on research design is available in the Nature Portfolio Reporting Summary linked to this article.

## Data availability

All the data used here are available from the Human Connectome Project[56] (http://www.humanconnectomeproject.org/) and the Microstructure-Informed Connectomics Project[57] (https://osf.io/j532r/). Data for reproducing figures Figs. 3, 5a, c is included Supplementary Data 1 and Supplementary Data 2, respectively.

## Code availability

MATLAB code for computing the TC, DTC, O-information, and S-information, as well as the simulated annealing, is attached to this manuscript as Supplementary Software 1.

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

## Acknowledgements

T.F.V. and M.P. are supported by the NSF-NRT grant 1735095, Interdisciplinary Training in Complex Networks and Systems. The funders had no role in study design, data collection and analysis, decision to publish, or preparation of the manuscript.

## Author contributions

T.F.V., M.E.P., and O.S. conceived of the project. T.F.V. and O.S. performed the formal, mathematical analysis. M.E.P. and O.S. analyzed the data. J.F. preprocessed and collated the fMRI data. T.F.V. and M.E.P. wrote the initial manuscript. O.S., J.F. provided editorial feedback. O.S. supervised the project.

## Competing interests

The authors declare no competing interests.
