## [Peer Review File · Communications Biology]

Reviewers' comments:

Reviewer #1 (Remarks to the Author):

The article by Varley and colleagues investigates both theoretical properties and practical applicability to fMRI data of the new information, a new tool to assess the balance between redundant and synergistic interactions in network datasets, allowing to establish whether subsets of signals are dominated by redundancy or synergy. The work contains both original results with relevance for the study of brain networks, and aspects that need to be better investigated or clarified.

The authors should consider that a spectral measure of O-information has been recently proposed to analyze rhythmic processes [R1]. This measure is related to the mutual information and O-information rates, which allow a dynamic analysis of multivariate time series. This measure should be discussed, also in view of the recent trends in neuroscience suggesting to analyze dynamic interactions in fMRI time series in contrast to "static" functional connectivity reflected by correlation or mutual information measures (see, e.g., [R2] for the importance of dynamic analysis, or [R3] for an application of dynamic analysis to higher-order interactions).

The correlation between TSE complexity and DTC or TC+DTC has been proved heuristically in the dataset analyzed for the study. To argue more strongly about this correlation, it should be determined on simulated datasets where the levels of synergy/redundancy can be controlled.

Related to the previous comment, the theoretical investigation of the properties of the O-information and its relationship with existing measures should be performed also in view of the recursive formulation of the O-information evidencing its gradient across orders, highlighted for instance in [R4] and [R1], which evidences how synergy and/or redundancy arise from mutual information measures computed adding a new variable to an existing subset of variables.

The authors should elaborate more on the use of global signal regression, which is a debated topic [R5]. Moreover, a discussion would be in order about the fact that blood flow and the global signal are present in different amounts across distinct brain systems [R6], and that blood flow, blood arrival time and respiration are central to the formation of the BOLD signal and are likely a font of redundancies and synergies unrelated to the neural information.

A main critical aspect of the analysis of fMRI data regards the evaluation of subsets of BOLD signals which actually exhibit statistically significant (positive or negative) values for the O-information. In the paper, it is not clear how many of the subsets associated with synergy present significant negative values of the O-information. Such statistical validation needs to be performed using e.g. surrogate data or bootstrap approaches.

Minor:

Eq. 1 provides the definition of entropy for discrete random variables, but then the analyses made under the Gaussian assumption implicitly refer to the differential entropy of continuous random variables.

The authors state that the interaction information is not addressed in this study (end of page 2), but actually this measure is implicitly accounted in the formulation of the O-information, since the O-information of three random variables coincides with the interaction information.

Please check whether the lowercase "n" in Eq. (8) has the same or different meaning than the uppercase "N" in Eq. (9).

[R1] Faes, L., Mijatovic, G., Antonacci, Y., Pernice, R., Barà, C., Sparacino, L., ... & Stramaglia, S. (2022). A Framework for the Time-and Frequency-Domain Assessment of High-Order Interactions in Brain and Physiological Networks. *IEEE Transactions on Signal Processing*, in press; arXiv preprint arXiv:2202.04179.

[R2] Novelli, L., & Razi, A. (2022). A mathematical perspective on edge-centric brain functional connectivity. *Nature communications*, 13(1), 1-13.

[R3] Luppi, A. I., Mediano, P. A., Rosas, F. E., Holland, N., Fryer, T. D., O'Brien, J. T., ... & Stamatakis, E. A. (2022). A synergistic core for human brain evolution and cognition. *Nature Neuroscience*, 25(6), 771-782.

[R4] Stramaglia, S., Scagliarini, T., Daniels, B. C., & Marinazzo, D. (2021). Quantifying dynamical high-order interdependencies from the o-information: an application to neural spiking dynamics. *Frontiers in Physiology*, 11, 595736.

[R5] Liu, T. T., Nalci, A., & Falahpour, M. (2017). The global signal in fMRI: Nuisance or Information?. *Neuroimage*, 150, 213-229.

[R6] Colenbier, N., Van de Steen, F., Uddin, L. Q., Poldrack, R. A., Calhoun, V. D., & Marinazzo, D. (2020). Disambiguating the role of blood flow and global signal with partial information decomposition. *Neuroimage*, 213, 116699.

Reviewer #2 (Remarks to the Author):

This is a very interesting paper, exploring high order statistical dependencies in brain signals, a very promising field which might lend new insights in the functioning of human brain. The manuscript is well written, in my opinion it contains material which deserves publication, after the revision of the following minor points:

- 1) the authors claim that global signal regression should remove redundancy, they should correctly cite the following paper <https://pubmed.ncbi.nlm.nih.gov/32179104/>
- 2) reference [17] has now been published: *Phys. Rev. Research* 4, 013184 – Published 4 March 2022
- 3) The authors should add a sentence where they state that a possible limitation of their findings might be the fact that the blood flow, blood arrival time, and respiration, which construct BOLD signals, might be responsible of part of the synergy and redundancy observed.

Reviewer #3 (Remarks to the Author):

This manuscript presents interesting new analyses of the information theoretic measure O-information, which is a measure of the balance between synergy and redundancy amongst a collection of variables. (Synergy being information that only arises from knowing 2 or more variables, and redundancy being information that is commonly held by more than one variable.) The theoretical analysis relates the O-information to the better-known Tononi-Sporns-Edelman complexity, and importantly gains some insight into the kind of synergy it is quantifying, namely the expected decrease in integration (total correlation) when one variable is removed. The measure is then applied to two resting state fMRI datasets to show that it is capable of picking up a bunch of higher-order structures in brain dynamics, i.e., statistical interactions that are occurring between groups of three or more variables, that are not reducible to interactions between pairs of variables. This opens up the opportunity for many new scientific insights into interactions between brain regions and their role in all

kinds of brain function.

I just have a few small points that I think it would be good to address.

The theoretical contribution on the kind of synergy O-information captures (that I summarised above) would be good emphasise in the abstract. At the moment the theoretical contribution is stated a bit vaguely in the abstract, and moreover, the sentence on this doesn't read well.

Although it is discussed in the discussion, it would be good to also briefly mention partial information decomposition (PID) in the introduction, possibly referring forward to the discussion, as otherwise the reader might be left wondering about it (as I was). I do like O-information as an alternative to PID, and more of the advantages could be stated. PID is not just computationally expensive, there is the fact that there is no unique redundancy function, and results can depend quite heavily on the choice of that.

(Top col 2, pg 2), it's not really a new idea to expand the notion of integration into different kinds, according to synergistic, unique and redundant transfers of information (see e.g., the reference below) but it's just the PID approaches haven't really gotten that far with empirical data because of the problems mentioned above. So, I think the contribution can be stated a bit more accurately here- the manuscript really shows how O-information is both theoretically sound and pragmatic.

Mediano, P.A.M., Rosas, F.E., Luppi, A.I., Carhart-Harris, R.L., Bor, D., Seth, A.K., & Barrett, A.B. (2021). Towards an extended taxonomy of information dynamics via Integrated Information Decomposition. arXiv 2109.13186.

Going further with this, on pg 8, where Luppi et al [21] is mentioned and it is found their results contrast somewhat with the present findings, another reason for this is the fact that their results relied on a choice of redundancy function, and may not hold for all possible reasonable redundancy functions.

It could be greater emphasised that this analysis is atemporal, and applies only to an assumed stationary probability distribution, and it would be good to speculate a bit about the best way forward for an analysis of dynamical information interactions, e.g., how best to extend transfer entropy to consider higher-order interactions, given that PID is problematic.

On pg2, where it is stated that the balance of segregated and integrated dynamics correlates with conscious awareness, only 2 very recent papers are cited. I think it's important to cite some older more seminal papers here, e.g.,

A.G. Casali, et al. A theoretically based index of consciousness independent of sensory processing and behavior Sci. Transl. Med., 5 (2013), 198ra105.

M. Massimini, et al. Breakdown of cortical effective connectivity during sleep Science, 309 (2005), 2228-2232.

And probably this recent review as well:

S. Sarasso, et al. Consciousness and complexity: a consilience of evidence Neurosci. Conscious. (2021)

(NB I'm not an author of any of these papers!)

Given that some very basic information theory is presented, it would be good to state (pg 3 col 1) what KL divergence is.

Similarly, above equation (16), it would be good to say that this is technically the differential entropy. It could even be mentioned that when you apply the discrete formula to continuous variables via binning, and then take the limit as the bin width goes to zero, the formula for entropy actually diverges, but if you ignore the term that diverges, you get the same expression for the mutual information as if you didn't ignore it. And the limit with the divergent term ignored is the differential entropy.

(Pg 9 col 1) What is the work by Sherill et al? Plus, the citation is missing.

Response to reviewers

We would like to thank the reviewers for taking the time to consider our manuscript. We have addressed all of the comments below and hope the reviewers feel that our paper is stronger now. Our replies are noted in blue font.

On behalf of all authors
- Thomas F. Varley

Reviewer 1 (Remarks to the Author):

The article by Varley and colleagues investigates both theoretical properties and practical applicability to fMRI data of the new information, a new tool to assess the balance between redundant and synergistic interactions in network datasets, allowing to establish whether subsets of signals are dominated by redundancy or synergy. The work contains both original results with relevance for the study of brain networks, and aspects that need to be better investigated or clarified.

The authors should consider that a spectral measure of O-information has been recently proposed to analyze rhythmic processes [R1]. This measure is related to the mutual information and O-information rates, which allow a dynamic analysis of multivariate time series. This measure should be discussed, also in view of the recent trends in neuroscience suggesting to analyze dynamic interactions in fMRI time series in contrast to “static” functional connectivity reflected by correlation or mutual information measures (see, e.g., [R2] for the importance of dynamic analysis, or [R3] for an application of dynamic analysis to higher-order interactions).

We thank the reviewer for their critical engagement with the literature. We have added the following to the discussion:

Another limitation is that it operates on “static” distributions: every frame is assumed to have been drawn from an unchanging multivariate Gaussian distribution, with no memory or dynamics from moment to moment. This is a standard assumption in functional connectivity analyses, although there is growing interest in the limitations this assumption produces and the need for analyses that explicitly account for dynamics (Novelli & Razi, 2021). The field of information dynamics provides a number of relevant analyses (Lizier et al., 2014; Bossomaier et al., 2016), and there is already interest in higher-order dynamics in the brain: in addition to the aforementioned work by Luppi et al., recent work by Faes et al. proposed a derivative of the O-information for rhythmic processes (the “O-information rate”, or OIR) (Faes et al., 2022). The OIR has been used to describe brain-heart interaction dynamics and opens up the frequency domain to higher-order, informational analysis in addition to the time domain. Similarly, an application of the O-information to the dynamic measure of transfer entropy has been proposed and applied to optimizing ensembles of maximally synergistic or redundant neurons (Stramaglia et al., 2021). Both of these measures could be incorporated into a pipeline like the one described here and may shed light on the similarities (and differences) between dynamic and static analyses. Despite its limitations, however, we are confident that the classic, static O-information likely contains a wealth of as-yet unexplored structure and will continue to provide insights into brain structure and function.

The correlation between TSE complexity and DTC or TC+DTC has been proved heuristically in the dataset analyzed for the study. To argue more strongly about this correlation, it should be determined on simulated datasets where the levels of synergy/redundancy can be controlled.

We strongly agree with the reviewer that this is an interesting question, well worth further study. We have added the following to the discussion:

The nature of this correlation merits further study as well. One outstanding question is how redundancies and synergies in the data differentially influence the correlation. Unlike the O-information, the S-information does not obviously link to redundancies or synergies, and so how these “kinds” of integration impact the relationship to TSE remains unknown. Future work developing generative models with precisely controllable distributions of redundancies and synergies may shed light on this question.

Related to the previous comment, the theoretical investigation of the properties of the O-information and its relationship with existing measures should be performed also in view of the recursive formulation of the O-information evidencing its gradient across orders, highlighted for instance in [R4] and [R1], which evidences how synergy and/or redundancy arise from mutual information measures computed adding a new variable to an existing subset of variables.

In response to the reviewer, we have explored how our analytic link between O-information and description complexity could provide insight into the gradients. Re-writing the partial O-information in terms of TCs is straightforward:

$$\partial_k \Omega(\mathbf{X}) = (2 - N)TC(\mathbf{X}) + \sum_{i=1}^N TC(\mathbf{X}^{-i}) - \left[(1 - N)TC(\mathbf{X}^{-k}) + \sum_{j=1, j \neq k}^N TC(\mathbf{X}^{-jk}) \right] \quad (1)$$

As it stands, this formulation is less than immediately illuminating. While there may be valuable links to explore here, doing it justice would require considerably more than can be done in the context of this revision. We have opted to make reference to the O-information gradient and suggest future avenues of work:

The framing of the O-information in terms of the change to integration after removal of individual elements also has conceptual links to the so-called “gradients of O-information” [?]. Scagliarini et al., explore how individual elements can contribute redundantly or synergistically to the O-information, defining the gradient as the difference between the O-information of an ensemble \mathbf{X} and the O-information when single elements X_i are excluded. While a detailed analytic exploration of the link is beyond the scope of this paper, the property of gradients yield valuable insights into the structure of higher-order dependencies in complex systems.

The authors should elaborate more on the use of global signal regression, which is a debated topic [R5]. Moreover, a discussion would be in order about the fact that blood flow and the global signal are present in different amounts across distinct brain systems [R6], and that blood flow, blood arrival time and respiration are central to the formation of the BOLD signal and are likely a font of redundancies and synergies unrelated to the neural information.

We have added the following to the Discussion section:

One limitation of this study is that it is hard to disambiguate between information that reflects “computation” in neural tissue, versus what is attributable to the vascular physiology of the BOLD signal. Recent work by Colenbier has shown that there are synergistic interactions between the global signal, blood arrival times, and functional connectivity structure (Colenbier, 2020). Since the pairwise covariance forms the foundation of the multivariate Gaussian entropy estimator, it is likely that the same confounds influence the estimates of entropy and mutual information. Future work replicating these results using electrophysiological recordings such as M/EEG should help untangle this issue.

A main critical aspect of the analysis of fMRI data regards the evaluation of subsets of BOLD signals which actually exhibit statistically significant (positive or negative) values for the O-information. In the

paper, it is not clear how many of the subsets associated with synergy present significant negative values of the O-information. Such statistical validation needs to be performed using e.g. surrogate data or bootstrap approaches.

We agree with the author that testing the validity of subsets is an important issue, and one that we should have expanded on. To address this, we have added the following analysis to the Results:

To validate that our optimization algorithm was observing “truly” synergistic ensembles, we tested each optimized subsystem against a null (see Materials and Methods E). Since the ensemble size k is fixed by the optimization algorithm, it is possible that the apparent synergy of that ensemble is actually due to some subset of nodes within that ensemble (for example, a system of three synergistic elements and two independent elements will appear to be a synergistic system of $k = 5$, however the “real” synergy is only in the three entangled elements. To ensure that all elements in the ensemble contributed to the synergy, we only considered a set “valid” if it was impossible to remove any node without the O-information increasing (i.e. the contribution of each element was synergy-dominated). We found that, for small value of k , the vast majority of optimized ensembles were valid ($\approx 99.08\%$ for ensembles of size four, $\approx 92.92\%$ for ensembles of size six, $\approx 84.14\%$ for ensembles of size eight, and $\approx 64.04\%$ for the maximally synergistic ensemble size of ten). For collections much larger than ten, the proportion of valid systems decreased rapidly: for ensembles of size fifteen, only $\approx 0.04\%$ were valid, and there were no valid ensembles of size greater than fifteen, despite the fact that the simulated annealing algorithm returned a large number of results with $\Omega < 0$. This implies that, although these larger subsystems *are* synergy dominated, that synergy is restricted to a “core” set of components and not attributable to the whole.

Minor:

Eq. 1 provides the definition of entropy for discrete random variables, but then the analyses made under the Gaussian assumption implicitly refer to the differential entropy of continuous random variables.

We have added the following elaboration to the Gaussian Information Theory section:

Since BOLD data is continuous (rather than discrete), to quantify the entropy of a continuous signal, we use a generalization of the the classic, discrete Shannon entropy (Eq. 1): the differential entropy:

$$H(X) = \int_{x \in \mathcal{X}} P(x) \log P(x) dx \quad (2)$$

Computing the differential entropy from empirical data is generally difficult, as it requires estimating $P(x)$. However, if one is willing to make assumptions of multivariate normality, closed-form estimators of the Gaussian joint entropy can be leveraged.

We have also added a brief reference to the differential entropy in in the initial introduction of Shannon entropy.

The authors state that the interaction information is not addressed in this study (end of page 2), but actually this measure is implicitly accounted in the formulation of the O-information , since the O-information of three random variables coincides with the interaction information.

We have added the following to the introduction of $\Omega(\mathbf{X})$:

In the specific case of three variables, $\Omega(X_1, X_2, X_3)$ is equivalent to the co-information [?], which Williams and Beer showed is itself equivalent to the redundancy minus the synergy [?]. This is in

keeping with the intuition that positive O-information implies a redundancy-dominated structure and a negative O-information implies a synergy-dominated structure, although the direct link between Ω and the co-information is only direct for three variables and the measures are not identical for larger sets.

Please check whether the lowercase “n” in Eq. (8) has the same or different meaning than the uppercase “N” in E1. (9)

Good catch, thank you.

[R1] Faes, L., Mijatovic, G., Antonacci, Y., Pernice, R., Barà, C., Sparacino, L., ...& Stramaglia, S. (2022). A Framework for the Time-and Frequency-Domain Assessment of High-Order Interactions in Brain and Physiological Networks. *IEEE Transactions on Signal Processing*, in press; arXiv preprint arXiv:2202.04179.

[R2] Novelli, L.,& Razi, A. (2022). A mathematical perspective on edge-centric brain functional connectivity. *Nature communications*, 13(1), 1-13.

[R3] Luppi, A. I., Mediano, P. A., Rosas, F. E., Holland, N., Fryer, T. D., O’Brien, J. T., ...& Stamatakis, E. A. (2022). A synergistic core for human brain evolution and cognition. *Nature Neuroscience*, 25(6), 771-782.

[R4] Stramaglia, S., Scagliarini, T., Daniels, B. C.,& Marinazzo, D. (2021). Quantifying dynamical high-order interdependencies from the o-information: an application to neural spiking dynamics. *Frontiers in Physiology*, 11, 595736.

[R5] Liu, T. T., Nalci, A.,& Falahpour, M. (2017). The global signal in fMRI: Nuisance or Information?. *Neuroimage*, 150, 213-229.

[R6] Colenbier, N., Van de Steen, F., Uddin, L. Q., Poldrack, R. A., Calhoun, V. D.,& Marinazzo, D. (2020). Disambiguating the role of blood flow and global signal with partial information decomposition. *Neuroimage*, 213, 116699.

We have added these relevant citations.

Reviewer 2 (Remarks to the Author):

This is a very interesting paper, exploring high order statistical dependencies in brain signals, a very promising field which might lead new insights in the functioning of human brain. The manuscript is well written, in my opinion it contains material which deserves publication, after the revision of the following minor points:

1) the authors claim that global signal regression should remove redundancy, they should correctly cite the following paper <https://pubmed.ncbi.nlm.nih.gov/32179104/>

We have updated the reference list with the Colenbier article.

2) reference [17] has now been published: *Phys. Rev. Research* 4, 013184 – Published 4 March 2022

We have updated the reference list with the correct version of the Scagliarini paper.

3) The authors should add a sentence where they state that a possible limitation of their findings might be the fact that the blood flow, blood arrival time, and respiration, which construct BOLD signals, might be responsible of part of the synergy and redundancy observed.

We thank the reviewer for this comment. Given its similarity a comment made by reviewer 1, we refer to the previous response. We have added the following to the Discussion section:

One limitation of this study is that it is hard to disambiguate between information that reflects “computation” in neural tissue, versus what is attributable to the vascular physiology of the BOLD signal. Recent work by Colenbier has shown that there are synergistic interactions between the global signal, blood arrival times, and functional connectivity structure. Since the pairwise covariance forms the foundation of the multivariate Gaussian entropy estimator, it is likely that the same confounds influence the estimates of entropy and mutual information. Future

work replicating these results using electrophysiological recordings such as M/EEG should help untangle this issue.

Reviewer 3 (Remarks to the Author):

This manuscript presents interesting new analyses of the information theoretic measure O-information, which is a measure of the balance between synergy and redundancy amongst a collection of variables. (Synergy being information that only arises from knowing 2 or more variables, and redundancy being information that is commonly held by more than one variable.) The theoretical analysis relates the O-information to the better-known Tononi-Sporns-Edelman complexity, and importantly gains some insight into the kind of synergy it is quantifying, namely the expected decrease in integration (total correlation) when one variable is removed. The measure is then applied to two resting state fMRI datasets to show that it is capable of picking up a bunch of higher-order structures in brain dynamics, i.e., statistical interactions that are occurring between groups of three or more variables, that are not reducible to interactions between pairs of variables. This opens up the opportunity for many new scientific insights into interactions between brain regions and their role in all kinds of brain function.

I just have a few small points that I think it would be good to address.

The theoretical contribution on the kind of synergy O-information captures (that I summarised above) would be good emphasise in the abstract. At the moment the theoretical contribution is stated a bit vaguely in the abstract, and moreover, the sentence on this doesn't read well.

We have added the following to the abstract:

We begin with a mathematical analysis of the O-information, showing analytically and numerically how it is related to previously established, information theoretic measures of complexity (such as the Tononi-Sporns-Edelman complexity and description complexity). This link allows us to propose a new, intuitive understanding of the O-information as quantifying the difference in integration between scales, and synergy as occurring when removal of any component element causes a greater decrease in integration than would be expected in a disintegrated system. We then apply the O-information to brain data, showing that synergistic subsystems are widespread in the human brain.

Although it is discussed in the discussion, it would be good to also briefly mention partial information decomposition (PID) in the introduction, possibly referring forward to the discussion, as otherwise the reader might be left wondering about it (as I was). I do like O-information as an alternative to PID, and more of the advantages could be stated. PID is not just computationally expensive, there is the fact that there is no unique redundancy function, and results can depend quite heavily on the choice of that.

See below for our added discussion of the PID.

(Top col 2, pg 2), it's not really a new idea to expand the notion of integration into different kinds, according to synergistic, unique and redundant transfers of information (see e.g., the reference below) but it's just the PID approaches haven't really gotten that far with empirical data because of the problems mentioned above. So, I think the contribution can be stated a bit more accurately here- the manuscript really shows how O-information is both theoretically sound and pragmatic.

We have added the following to the introduction:

Much of the previous work on higher-order information in neuroscience has used the *partial information decomposition* (PID) framework (Williams & Beer, 2010; Gutknecht et al., 2021), which provides a complete decomposition of the joint mutual information to "atomic" information components. While powerful, the PID framework has some fairly strict limitations that have hindered it's adoption by the wider complex systems community. The first is that it requires

partitioning a system into “sources” and “targets”, and does not allow analysis of the whole system as itself. The second is that, due to the combinatorial explosion of information atoms, analysis of more than five or six elements is impossible. Given that even small systems can have hundreds, or even thousands of elements, this is a severe limitation. Finally, the PID is unusual in that, while it reveals the *structure* of multivariate information, actually calculating values from data requires an additional step: the selection of a redundancy function that quantifies some notion of “redundant information.” This is a surprisingly difficult task, as many redundancy functions have been proposed, and different choices can lead to radically different descriptions of the same system (Kolchinsky 2022; Kay et al., 2022)

Furthermore, we have re-written the sentence where the O-information is introduced to more explicitly contrast it to the PID framework.

Mediano, P.A.M., Rosas, F.E., Luppi, A.I., Carhart-Harris, R.L., Bor, D., Seth, A.K., & Barrett, A.B. (2021). Towards an extended taxonomy of information dynamics via Integrated Information Decomposition. arXiv 2109.13186.

Going further with this, on pg 8, where Luppi et al [21] is mentioned and it is found their results contrast somewhat with the present findings, another reason for this is the fact that their results relied on a choice of redundancy function, and may not hold for all possible reasonable redundancy functions.

We have added the following to our discussion of Luppi et al., (2022):

Finally, the prior analysis is based on a generalization of the partial information decomposition and requires choosing one of several redundancy functions. It is unknown whether the reported results would hold for all plausible redundant information functions.

It could be greater emphasised that this analysis is atemporal, and applies only to an assumed stationary probability distribution, and it would be good to speculate a bit about the best way forward for an analysis of dynamical information interactions, e.g., how best to extend transfer entropy to consider higher-order interactions, given that PID is problematic.

See our response to a very similar comment from Reviewer 1.

On pg2, where it is stated that the balance of segregated and integrated dynamics correlates with conscious awareness, only 2 very recent papers are cited. I think it’s important to cite some older more seminal papers here, e.g.,

A.G. Casali, et al. A theoretically based index of consciousness independent of sensory processing and behavior *Sci. Transl. Med.*, 5 (2013), 198ra105.

M. Massimini, et al. Breakdown of cortical effective connectivity during sleep *Science*, 309 (2005), 2228-2232.

And probably this recent review as well:

S. Sarasso, et al. Consciousness and complexity: a consilience of evidence *Neurosci. Conscious.* (2021) (NB I’m not an author of any of these papers!)

I have added all three of those citations to the relevant section.

Given that some very basic information theory is presented, it would be good to state (pg 3 col 1) what KL divergence is.

We have added that to our basic information theory section.

Similarly, above equation (16), it would be good to say that this is technically the differential entropy. It could even be mentioned that when you apply the discrete formula to continuous variables via binning, and then take the limit as the bin width goes to zero, the formula for entropy actually diverges, but if you ignore the term that diverges, you get the same expression for the mutual information as if you didn’t ignore it. And the limit with the divergent term ignored is the differential entropy.

We have added the following elaboration to the Gaussian Information Theory section:

Since BOLD data is continuous (rather than discrete). To quantify the entropy of a continuous signal, we use a generalization of the the classic, discrete Shannon entropy (Eq. ??): the differential entropy:

$$H(X) = \int_{x \in \mathcal{X}} P(x) \log P(x) dx \quad (3)$$

Computing the differential entropy from empirical data is generally difficult, as it requires estimating $P(x)$. However, if one is willing to make assumptions of multivariate normality, closed-form estimators of the Gaussian joint entropy can be leveraged.

While the discussion of diverging estimators is interesting, we feel that it is slightly beyond the scope of this paper, and have opted for a more succinct introduction of the differential entropy.

(Pg 9 col 1) What is the work by Sherill et al? Plus, the citation is missing.

We have added to the section more explicitly explaining the approach that Sherill et al., took (a dynamic one) and contrasted it with the atemporal nature of the O-information.

REVIEWERS' COMMENTS:

Reviewer #3 (Remarks to the Author):

My comments have been mostly well-addressed, and the manuscript has been improved.

I just have a couple of small comments on the revisions:

The edited part of the abstract is a useful addition, but is not well-written. I think the authors should have another attempt at it, seeking greater clarity, probably aiming for shorter sentences.

(pg 2 col 1) "Incredibly computational cost" – replace with something more scientific like "super-exponential computational cost".

There are some typos and instances of inaccurate grammar, but I assume these will come out during the final edits.

Reviewer #4 (Remarks to the Author):

Major comment 1: I believe the authors have addressed this comment properly. I would also suggest that the following reference to be added and briefly discussed in relation to the concepts of the manuscript:

Santoro, A., Battiston, F., Petri, G. et al. Higher-order organization of multivariate time series. *Nat. Phys.* 19, 221–229 (2023).

Major comment 2: I agree with Reviewer 1 that simulation, in addition to the heuristic evaluation of Figure 2, would substantially aid in demonstrating the relationship between TSE, DTC, and TC in the paper.

Major comment 3: I concur with the authors that theoretical research into the O-information gradient is likely outside the purview of this work and *Communications Biology* as a whole.

Major comment 4: I concur with Reviewer 1 that the use of global signal regression in functional connectivity research is controversial and that it has the potential to contribute erroneous negative correlations to the functional connectome. Therefore, the authors could possibly also include the analysis results without applying global signal regression in addition to their current results.

Major comment 5: The response by the authors seems sufficient to me.

Reviewer #5 (Remarks to the Author):

The authors have satisfactorily addressed the comments to the reviewers. I highly recommend thoroughly proofreading the manuscript since there are many typos and careless language errors, especially in the newly added text.